Relationship between higher education teachers’ affect and their psychological adjustment to online teaching during the COVID-19 pandemic: an application of latent profile analysis

Zou Weixing 1 2
Ding Xiangmei 1
Xie Lingping 1
Wang Hongli hongliw@xynun.edu.cn 1 2
1 Xingyi Normal University for Nationalities , Xingyi , China
2 School of Psychology, Guizhou Normal University , Guiyang , China
Mitsouras Katherine
Electronic publication date: 2021 Nov 3
Publication date: 2021
Volume: 9
Electronic Location ID: e12432
Received 2021 Jun 4; Accepted 2021 Oct 13
Copyright: ©2021 Zou et al.
Copyright year: 2021
Copyright holder: Zou et al.
License: This is an open access article distributed under the terms of the Creative Commons Attribution License, which permits unrestricted use, distribution, reproduction and adaptation in any medium and for any purpose provided that it is properly attributed. For attribution, the original author(s), title, publication source (PeerJ) and either DOI or URL of the article must be cited.
License URL: https://creativecommons.org/licenses/by/4.0/

Keywords: Positive affect, Negative affect, Psychological adjustment, Online teaching, Latent profile analysis, COVID-19 pandemic

Funding: The Discipline Construction Fund of Xingyi Normal University for Nationalities Southwest Guizhou Autonomous Prefecture Technology Planning Project This research was funded by the Discipline Construction Fund of Xingyi Normal University for Nationalities, Southwest Guizhou Autonomous Prefecture Technology Planning Project. The funders had no role in study design, data collection and analysis, decision to publish, or preparation of the manuscript.

==============================
Background

The COVID-19 outbreak has forced teachers to transition to online teaching, requiring them to adapt their courses and pedagogical methods to an online format rapidly without relevant training. This has presented a formidable challenge to higher education teachers. The present study uses a person-centered approach to identify heterogeneity among higher education teachers’ affective experiences and the relationship between this heterogeneity and their psychological adjustment to online teaching.

Methods

In total, 2,104 teachers in higher education institutions in Southern China were surveyed using the Positive and Negative Affect Schedule and the Psychological Adjustment to Online Teaching Scale (a measure developed for this study) between March 25 and March 31, 2020. The collected data were analyzed using latent profile analysis.

Results

Based on their affective experiences during online teaching immediately after the COVID-19 outbreak, higher education teachers were divided into three latent classes: the common, ambivalent, and positive types. Among them, the positive type accounted for the largest proportion (44.85%), while the ambivalent type accounted for the smallest proportion (23.93%). The rest was the common type, which accounted for 31.15%. Significant differences in psychological adjustment to online teaching were found between the three latent classes. Regarding positive psychological adjustment, teachers belonging to the ambivalent type had significantly lower scores than those belonging to the other two types. Further, the common type had a significantly lower score than the positive type. Regarding negative psychological adjustment, the ambivalent type had a significantly higher score than the other two types, and the common type had a significantly higher score than the positive type.

Conclusion

Based on a novel person-centered perspective, this study revealed the differences and complexity in higher education teachers’ affective experiences of online teaching immediately after the COVID-19 outbreak. The three different types of affective experiences (common, ambivalent, and positive) had a significant influence on psychological adjustment, with teachers belonging to the ambivalent type showing the worst psychological adjustment. This study provides a new perspective for the discussion of the relationship between teachers’ affective experiences and their psychological adjustment to online teaching.

Introduction

The worldwide novel coronavirus outbreak has had a significant impact on all aspects of daily life. It has caused higher education institutions to transition to remote learning. On March 11, 2020, the World Health Organization designated COVID-19 as a global pandemic (World Health Organization, 2020). Regarding its impact on education, by the end of June 2020, over one billion learners were affected by the closure of their schools, colleges, and universities due to the COVID-19 outbreak (UNESCO, 2020a; UNESCO, 2020b). To address this health emergency, conventional classroom education was immediately moved online (Bozkurt & Sharma, 2020). This global education crisis forced teachers to adjust their curriculum teaching methods quickly without first receiving training on online teaching and digital teaching technology. Teachers in elementary schools, middle schools, and universities all felt this challenge (Ali, 2020; Gelles et al., 2020).

Psychological adjustment

Psychological adjustment refers to individuals’ active response to changes in the external environment to bring their psychological and behavioral patterns in line with the environmental changes and their developmental needs, thereby striking a balance between themselves and their environment (Cheng, Lau & Chan, 2014). Previous studies have used indicators such as loneliness, depression, anxiety, and self-esteem to reflect psychological adjustment (Chen, Wang & Liu, 2012; Chen et al., 2013; Coplan et al., 2017).

Lazarus’ transactional theory of stress and coping provides a theoretical framework for how individuals maintain psychological well-being even under the stress of severe illness, in addition to explaining individual performance differences (Walker, Jackson & Littlejohn, 2004). According to this theory, individuals’ resources, situational attribution, cognitive evaluation, and coping strategies are some of the important factors affecting their adjustment outcome. In addition, the holistic person-context interaction theory suggests that individuals ultimately achieve positive psychological adjustment in two ways: inner interactions (those among the physiological, psychological, and behavioral aspects of an individual) and outer interactions (the frequent interactions between individuals and their environment) (Magnusson & Stattin, 1998). Consequently, teachers’ affective states during the pandemic would be closely related to their psychological adjustment to online teaching. Moreover, for further adaptation of online teaching or blended learning, the changes in China’s higher education in response to the pandemic must be determined.

It should be noted that, in this paper, teachers’ psychological adjustment to online teaching specifically refers to their adjustment to online teaching that started after China initiated level-1 response measures for a major public health emergency. These measures included an immediate shutdown of in-person classes and a sudden shift to online learning. Quarantine and confinement necessitated the search for innovative approaches to teaching and learning assessment (Ripoll, Godino-Ojer & Calzada, 2021). Under these circumstances, higher education teachers had to balance their psychological and behavioral patterns with the new teaching mode and environment.

Positive affect, negative affect, and psychological adjustment

In 1988, Watson and colleagues developed the two-factor model of affect, classifying it into positive affect (PA) and negative affect (NA), which are independent dimensions that can be either high or low. PA reflects the extent to which people feel enthusiastic, active, or alert. High PA is a state of energetic, full attention, and willingness to engage, while low PA is a state of sorrow and loss of power. NA reflects the extent to which a person feels nervous, fearful, angry, and guilty. High NA is a state characterized by a negative attitude toward oneself and the world, while low NA is “a state of calmness and serenity” (Watson, Clark & Tellegen, 1988). Generally, individuals with more PA and fewer NA experiences are more likely to perceive their lives as happy (Diener et al., 1999). Following the COVID-19 outbreak, there were considerable changes in the social network environment. The impossibility of resuming in-person classroom teaching could have substantially influenced teachers’ affective states.

Smith & Lazarus (1990) proposed that emotions represent a class of solutions to adaptive problems, with each emotion representing an individual’s appraisal of the person–environment relationship. Similarly, several recent empirical studies (Mayer, Caruso & Salovey, 2016; Duchesne, Ratelle & Feng, 2017) have also directly or indirectly demonstrated that emotion plays a vital role in adaptive problems. According to Fredrickson’s broaden-and-build theory, PA broadens one’s awareness and scope of knowledge and encourages flexible and creative thinking. Meanwhile, NA leads to a defensive state, narrowing one’s scope of thinking and knowledge (Fredrickson, 2001). Broadening one’s cognitive horizon and positive attitudes can facilitate the process of coping with stress, helping build lasting psychosocial resources over time (Gloria, Faulk & Steinhardt, 2013). These resources contribute to enhancing individual resilience (Fredrikson et al. 2003). In addition, previous studies have found that positive emotions directly affect individuals’ psychological functions (Nelson & Knight, 2010), physical fitness (Fredrickson & Levenson, 1998), and stress coping capabilities (Gloria, Faulk & Steinhardt, 2013). Regarding negative emotions, studies have found that they are closely associated with adverse effects such as anger, anxiety, and depression (Ng et al., 2019), and they could lead to aggressive behaviors (Donahue et al., 2014). These previous studies partly support the broaden-and-build theory and suggest that a positive affective state and good psychological functioning are conducive to individual adaptability. However, using path analysis, the study found that both PA and NA can influence depression; however, only PA can effectively relieve depression (Kratz, Ehde & Bombardier, 2014). Therefore, considering the influence of PA or NA on the individual’s mental state alone, previous studies have inconsistent conclusions, and the influence mechanism of PA and NA on the individual’s psychological development needs to be explored in depth.

Affective states and work performance

Furthermore, when dealing with complex work requirements, people often show distinctive emotional responses, such as PA, NA, or other complex affective states, which influence their organizational performance and creativity (George & Zhou, 2002; George & Zhou, 2007) or lead to deviant behavior (Dalal et al., 2009). However, regarding employee creativity, the relationship between PA or NA and creativity may vary. According to the broaden-and-build theory (Fredrickson, 2001), PA can expand individuals’ thinking space and scope of knowledge in the short term, which can enhance the flexibility of their thinking, thereby improving employee creativity (Isen et al., 1985; Davis, 2009). Conversely, from the affect-as-information perspective, the good information provided by PA can lead to the perception of an enjoyable environment, which may be interpreted as a sign that one can relax and hard work is unnecessary. In such a case, people tend to rely only on common procedures and prior knowledge structures to cope with their work. This tendency is not conducive to high-level creativity at work (George & Zhou, 2002).

Regarding NA as well, there is no consistent conclusion on its impact on employee creativity. On the one hand, some previous studies have suggested that NA is detrimental to creativity at work (Isen, Daubman & Nowicki, 1987), as it could narrow individuals’ scope of knowledge and thinking space. On the other hand, some other studies have demonstrated that if individuals have a positive interpretation of the information obtained from NA, then it could be conducive to their creativity at work (George & Zhou, 2002). Prior studies have discussed the positive effects of NA on creativity from different theoretical perspectives. For example, based on the affect-as-information perspective, George & Zhou (2007) suggested that NA implies that someone is in an unfriendly environment, which could motivate them to find systematic or bottom-up solutions for their problems. Although this process may be time-consuming and knowledge-demanding, it pushes individuals to identify better methods to address their problems or tackle them by thinking more creatively.

Teachers’ psychological states during the COVID-19 pandemic

Globally, teachers are faced with high work pressure and mental problems. A research review pointed out that teachers’ work pressure is an important cause of their mental health problems (Naghieh et al., 2015). China’s frequent educational reforms to adapt to its rapid economic development have had a negative impact on the mental health of teachers. At the same time, the results of meta-analysis show that the mental health of university teachers is worse than that of primary and secondary school teachers (Yang et al., 2019). The professional characteristics of university teachers are mainly manifested in not only attending classes, preparing lessons, correcting homework, guiding students in internships and graduation thesis, but also in needing to spend time studying and charging, writing scientific research papers, etc. The particularity of the work of university teachers and the high degree of work pressure make them a high-risk group in terms of mental health (Chen et al., 2014). As the backbone of cultivating high-level talents for the country, college teachers’ mental health statuses will not only affect the average mental health of Chinese people, but will also affect the quality of higher education to a large extent. Therefore, it is necessary and realistic to study the mental health of university teachers and to propose corresponding preventive measures.

Although China’s first large-scale online teaching program has achieved and even exceeded the expected goal and can be described as successful, it has also had a profound impact on higher education teachers who were caught unawares by many of the developments in higher education (Wu & Li, 2020). However, most studies on the program’s impact have focused on the psychological states of students and the general public (Husky, Kovess-Masfety & Swendsen, 2020; Li et al., 2020a; Li et al., 2020b; Torales et al., 2020; Yi, Xi & Xue, 2020; Li et al., 2021). Only a few have addressed teachers’ psychological experiences. Some studies have focused on the anxiety and pressure of Chinese teachers during the COVID-19 pandemic (Li et al., 2020a; Li et al., 2020b; Zhou & Yao, 2020), but no research has focused on the psychological changes of higher education teachers in the early stages of online teaching. However, due to the uniqueness of the professional courses it teaches, the education and teaching work of higher education teachers is very special. College teachers obtained significantly higher pathological prevalence than the other two (i.e., kindergarteners and teachers of middle or primary schools: Zhang & Lu, 2008). Online teaching is a challenging and novel pedagogical mode, at the same time, many problems have been encountered in the effective development of teaching work for higher education teachers (Johnson, Veletsianos & Seaman, 2020).

Under the unprecedented COVID-19 pandemic, higher education teachers have likely undergone complex psychological experiences. They need to engage in creative thinking and learning, and effective implementation of teaching strategies to adapt to the new and demanding task of online teaching (Wu & Li, 2020). The present study thus aims to examine the experiences of teachers in higher education institutions, in terms of their psychological state and adjustment, immediately after the start of online teaching.

In the extraordinary period following the COVID-19 outbreak when people worked together to fight the pandemic by avoiding non-essential outings, higher education teachers who faced the novel and challenging task of online teaching may have undergone PA and NA experiences. Examining the relationship between teachers’ experiences and their pedagogical and psychological adjustment would be an interesting endeavor. The inconsistent results of previous studies on the effects of PA and NA may be attributable to the research methodology. Most of these studies adopted the variable-centered approach to examine the relationship of PA and NA with other variables separately. For example, a common approach has been to measure various PA experiences to calculate a total PA score by summing the component scores, without focusing on a specific PA or distinguishing between different PA experiences (Ramsey & Gentzler, 2015). This may be the reason for the inconsistency between previous studies. However, some meaningful and significant insights not found in previous studies can be gained by examining how individuals develop different specific PA, for example, through the interaction between emotions and other psychological processes such as cognition, or the spontaneous mechanisms of the emotions themselves.

Present study

In contrast, latent profile analysis (LPA) is a statistical analysis method based on the person-centered approach. It assumes the possibility of classifying a group of people to analyze the unique characteristics of each class. Researchers can use this approach to understand how different variables are combined and to attribute the outcome to specific groups. In other words, different types of subgroups can be identified based on the property and extent of the explicit variables, which can help examine within-group heterogeneity that cannot be observed in variable-centered studies (Wang & Hanges, 2011).

In the present study, we aimed to examine higher education teachers’ affective states during online teaching after the COVID-19 outbreak and their relevant psychological adjustment. We separately used both PA and NA indicators as prior studies have demonstrated that PA and NA do not necessarily have a completely negative relationship but instead tend to change independently (Deng et al., 2016). Therefore, examining the PA and NA indicators separately may be more effective.

In summary, there are no previous person-centered studies on the relationship between higher education teachers’ affective experiences and their psychological adjustment. Immediately after the outbreak of COVID-19, higher education teachers had to transition to online teaching. Facing a new teaching mode, they may have experienced complex emotions or a combination of different emotions. The effects of these emotions on their psychological adjustment to online teaching can reflect the roles of diverse emotions. Prior studies have separately examined the effects of a certain emotion (such as PA or NA) on psychological state. This approach may be responsible for the abovementioned inconsistencies in their conclusions. The purpose of this study was to use LPA to explore the characteristics of heterogeneity and different types of higher education teachers’ subgroups and proportions based on their affective experiences during the period immediately after the COVID-19 outbreak. In addition, it aimed to identify the relationship between teachers’ affective experiences and their psychological adjustment to online teaching. The following two hypotheses were proposed:

H1. There are at least two latent classes of higher education teachers based on their affective experiences during online teaching immediately after the COVID-19 outbreak.

H2. Teachers belonging to different classes show significant differences in their scores in psychological adjustment to online teaching.

Methods

Participants and procedure

This study adopted a convenient sampling method using the Questionnaire Star online survey platform (https://www.wjx.cn). First, the research team contacted many universities in southern China to request assistance in the investigation. Then, the teaching management department of each university was requested to forward the link of the questionnaire to all university teachers. In total, 2,197 teachers at higher education institutions in several provinces in Southern China were recruited. An online questionnaire survey was administered to them between March 25 and March 31, 2020 (about two weeks after most universities in Southern China had started online teaching). They needed approximately 10 min to complete the questionnaire. To avoid the possibility of missing data, questionnaires were set up so that empty responses could not be submitted.

Further, excessively short questionnaire response times have been deemed a valid reason for rejecting responses (Leiner, 2019). After eliminating questionnaires that were answered regularly and those answered in less than 1 min, 2,104 valid questionnaires were finally obtained, with an effective rate of 95.8%. Among the respondents, 878 were men and 1,226 were women. There were 278 professors, 708 associate professors, 769 lecturers, and 349 teaching assistants. Regarding subjects, 1,058 taught social sciences, 729 taught science and technology, and 317 taught art and sports. Their ages were between 20 and 69 years, and they had a teaching experience of 0.5–40 years. All participants were required to provide electronic informed consent prior to participation. Their participation was voluntary and responses were anonymous. Ethical approval for the study was obtained from the Research Ethics Committee of the School of Education Science, Xingyi Normal University for Nationalities (No. ESXYNUN20200315). All procedures were in accordance with the 1964 Helsinki declaration and its later amendments or comparable ethical standards.

Instruments

Positive and negative affect schedule (PANAS)

This study used the PANAS developed by Watson, Clark & Tellegen (1988). Moreover, we utilized the Chinese version of the scale translated and revised by Huang, Yang & Li (2003). This tool has good reliability and validity, is suitable for Chinese people, and is widely used in China (Fang et al., 2020; Jiang et al., 2019). The scale consists of experience descriptors for PA (such as interested, proud, and inspired) and NA (such as nervous, upset, and ashamed). Participants were required to indicate to what extent they experienced the emotions in the previous week with a 5-point scale (1 = very slightly or not at all, 2 = a little, 3 = moderately, 4 = quite a bit, 5 = extremely). In this study, the α reliability of the PA subscale was 0.926 and that of the NA subscale was 0.876.

The psychological adjustment to online teaching scale for teachers in higher education institutions

We developed the Psychological Adjustment to Online Teaching Scale specifically for this study. First, we conducted one-to-one virtual interviews with 66 teachers with different seniority levels working in more than 20 higher education institutions in Southwest China. The interview questions covered online teaching-related discomfort, pressure, worries, anxieties, complaints, and difficulties, in addition to positive emotions for the novel online teaching mode. Detailed records of the interviews were maintained to develop structured questions for the scale. Based on the records and possible difficulties of online teaching, as well as on anxiety and depression scales, we developed a draft of the Psychological Adjustment to Online Teaching Scale, which covered the two dimensions of positive adjustment and negative adjustment. The draft included 20 items in total. Subsequently, a panel of five experts in psychology and pedagogy discussed the draft’s structure and all items. We finalized the scale after excluding controversial items, merging similar or repetitive ones, and removing or revising those that were difficult to understand. The final scale is concise and can be completed quickly. It was instrumental in obtaining reliable data. It consists of six positive adjustment items (such as “During the online network teaching, I actively share and exchange my experience with other teachers to support and help each other”) and eight negative adjustment items (e.g., “When teaching online, various network failures (jam, dropped line, etc.) make me confused, frustrated and upset.”), with 14 items in total.

To test the reliability and validity of the scale, we split the valid data set obtained from the survey with the final scale into odd and even parts. The odd data were used for exploratory factor analysis (EFA), and the even data were used for confirmatory factor analysis (CFA). Specifically, EFA was conducted on 14 items with odd data. The Kaiser–Meyer–Olkin (KMO) value was 0.859, and Bartlett’s sphericity test was statistically significant (χ2/df = 4182.952/91, p < 0.001), indicating the suitability of the items for factor analysis. Subsequently, principal component analysis was used to extract common factors, and maximum rotation was performed. The factor loadings are shown in Table 1.

Table 1 Factor loadings of principal component analysis.

Scale items	Factor 1	Factor 2	
N4	0.762		
N5	0.727		
N2	0.724		
N8	0.720		
N3	0.706		
N1	0.667		
N6	0.605		
N7	0.471		
P3		0.757	
P4		0.706	
P1		0.682	
P2		0.577	
P6		0.573	
P5		0.524	
Eigenvalues	4.329	2.163	
Contribution rates (%)	30.922%	15.450%	
Notes.

P1-6 positive adjustment items

N1-8 negative adjustment items

Table 1 shows that EFA generated two common factors. The cumulative variance explanatory rate after rotation was 46.372%. Factor analysis results were thus consistent with theoretical expectations. Factor 1 contains all negative adjustment items, and Factor 2 contains all positive adjustment items, indicating that the scale has good structural validity.

Meanwhile, the even data were used to conduct CFA for the confirmed scale structure using the structural equation modeling software AMOS 20.0. The results showed that RMSEA = 0.081, CFI = 0.878, NFI = 0.864, IFI = 0.879, GFI = 0.922, and χ2/df = 7.942. All fit indexes were acceptable, and the structure of the surface model was generally in compliance with survey requirements.

The reliability test of the scale showed that the α reliability coefficient of the positive adjustment dimension was 0.737 and that of the negative adjustment dimension was 0.835, indicating a high consistency of the two dimensions.

Data analysis

MPLUS 7.4 was used to analyze the latent profiles of PA and NA to judge their latent classes and distribution. SPSS 20.0 was used for variance analysis to explore the relationship between the latent classes of higher education teachers based on their affective states and their psychological adjustment to online teaching. When analyzing the profile of teachers’ affective states, the number of profiles in the model was gradually increased from a two-profile model to find the best fitting model. The fit indexes were mainly based on the following (Foti et al., 2012): (1) Akaike information criteria (AIC), Bayesian information criteria (BIC), and sample-size adjusted BIC (SSA-BIC), with smaller information index indicating a better fit; (2) entropy, which was used to evaluate the accuracy of classification, with larger entropy indicating more accurate classification; and (3) Lo-Mendell-Rubin (LMR) likelihood ratio test and Bootstrap likelihood ratio test (BLRT), which were used to compare two nested models, with significant p-values indicating that the k-class model had a better fit than the k-1-class model (Clark, 2010).

Results

Common method deviation test

This study first used Harman’s single-factor test to analyze the common method deviation (Podsakoff et al., 2003). The EFA results showed that the variation explained by the first factor without rotational precipitation was 30.837%, less than the critical value of 40%. In addition, CFA showed that the fitting results of the single-factor model were not good (χ2/df = 55.465, RMSEA = 0.161, CFI = 0.467, TLI = 0.416, SRMR = 0.128). Therefore, in this study, the common method deviation did not cause serious issues.

Descriptive statistics

The mean values, standard deviations, and correlation coefficients of the study variables are shown in Table 2. Among them, PA was significantly and positively correlated with positive adjustment and significantly and negatively correlated with negative adjustment. In contrast, NA was significantly and negatively correlated with positive adjustment and significantly and positively correlated with negative adjustment.

Table 2 Mean, standard deviation, and correlation coefficients of the positive adjustment, negative adjustment, PA, and NA.

	M ± SD	Positive adjustment	Negative adjustment	PA	NA	
Positive adjustment	3.613 ± 0.620	1				
Negative adjustment	2.504 ± 0.684	−0.258***	1			
PA	3.389 ± 0.910	0.524***	−0.273***	1		
NA	1.629 ± 0.679	−0.353***	0.548***	−0.277***	1	
Notes.

*** p < 0.001.

PA positive affect

NA negative affect

LPA results

LPA results are shown in Table 3. The model indexes indicated that, with an increase in the number of model classes, the fit values of AIC, BIC, and SSA-BIC decreased the most and declined gently in Model 3. Compared with Model 2, both the LMR-LRT and BLRT of Model 3 were significant. Moreover, the entropy value of Model 3 was higher than that of Model 2 (0.874 > 0.859). Further, compared with Model 3, the LMR-LRT values of Models 4 and 5 were not significant. Therefore, Models 4 and 5 showed no effective improvement over Model 3. Considering the simplicity and accuracy of the models, we finally selected Model 3 as the best fitting model.

Table 3 Summary of latent profile model fit indexes of teachers’ affective experiences.

Model	AIC	BIC	SSA-BIC	Entropy	LMR-LRT	BLRT	Share of each class	
1	56386.293	56499.325	56435.783	–	–	–	–	
2	51211.955	51387.154	51288.664	0.859	<0.001	<0.001	39.00/61.00	
3	48034.755	48272.122	48138.682	0.874	<0.001	<0.001	31.15/23.99/44.85	
4	45925.273	46224.808	46056.421	0.895	0.360	0.357	37.92/7.96/35.03/19.08	
5	44495.821	44857.523	44654.188	0.922	0.166	0.166	31.44/22.28/8.20/29.92/8.16	
Notes.

AIC Akaike information criteria

BIC Bayesian information criteria

SSA-BIC sample size adjusted BIC

LMR-LRT Lo-Mendell-Rubin likelihood ratio test

BLRT Bootstrap LRT

The scores of the three latent profile classes on the PA and NA dimensions are shown in Fig. 1. Profile 1 contained 31.15% of the participants. The PA experience of the teachers in this profile was lower than that of those in the other two profiles; their NA experience was also relatively low. Therefore, this profile was named the common type. Further, Profile 2 contained only 23.99% of the participants. The teachers in this profile had comparable scores in the two dimensions, which were higher than those of the teachers in Profile 1. Therefore, Profile 2 was named the ambivalent type. Lastly, the teachers in Profile 3 (44.85%) had significantly higher PA scores than those in the other two profiles; they also had the lowest NA scores. Therefore, Profile 3 was named the positive type.

Figure 1 Average positive affect (PA) and negative affect (NA) scores of the latent classes of higher education teachers based on their affective states.

Relationship between the latent classes based on higher education teachers’ affective states and their psychological adjustment to online teaching

One-way analysis of variance was used to explore the relationship between the latent classes and teachers’ psychological adjustment to online teaching (Table 4). The results showed significant differences in the scores of the latent classes on the positive adjustment dimension (F[2, 2101] = 328.488, ηp2 = 0.238, p < 0.001). The post-hoc test using the method of LSD showed that the positive adjustment score of the ambivalent type was significantly lower than those of the positive (p < 0.001) and common types (p < 0.01), and that of the common type was significantly lower than that of the positive type (p < 0.001). Regarding negative adjustment to online teaching, significant differences were found in the scores of the three latent classes on the negative adjustment dimension as well (F[2, 2101] = 295.155, ηp2 = 0.219, p < 0.001). The post-hoc test showed that the negative adjustment score of the ambivalent type was significantly higher than those of the positive and common types (ps < 0.001), and that of the common type was significantly higher than that of the positive type (p < 0.001).

Table 4 One-way analysis of variance of the relationship between higher education teachers’ psychological adjustment to online teaching and the three latent profile classes based on the teachers’ affective experiences.

	Common type
(Profile 1)	Ambivalent type
(Profile 2)	Positive type
(Profile 3)	F	Post hoc
(LSD)	
Positive adjustment	3.387 ± 0.592	3.287 ± 0.538	3.945 ± 0.506	328.488***	2 < 1, 3
1 < 3	
Negative adjustment	2.488 ± 0.628	3.306 ± 0.605	2.231 ± 0.589	295.155***	2 > 1, 3
1 > 3	

Discussion

Potential categories of higher education teachers’ online teaching affective experiences after the COVID-19 outbreak

In this study, from a person-centered perspective, we used LPA to ascertain the latent classes of higher education teachers based on their affective states in relation to online teaching after the COVID-19 outbreak. We identified three obviously latent classes and named them the common, ambivalent, and positive types. Among them, teachers within the positive type group accounted for the highest proportion, while those within the ambivalent type group accounted for the lowest proportion. The three latent profiles based on teachers’ affective states helped confirm H1. Teachers within the positive type group had the highest positive affective experience and the lowest negative affective experience. On the other hand, the positive affective experience and negative affective experience of the teachers within the ambivalent type group were similar, showing that the affective state is bad. The affective experience of common type teachers was at a normal state. During the period immediately after the COVID-19 outbreak, higher education teachers experienced complex affective states. UNESCO has also confirmed that confusion and pressure experienced by teachers is one of the negative consequences of school closures because these measures are too sudden, the duration is uncertain, and the teachers often are unfamiliar with distance online education and teaching (UNESCO, 2020a; UNESCO, 2020b). We also found considerable heterogeneity among individuals, which supports H2. The identified classes are partly in line with the results of previous studies (Deng et al., 2016), and they demonstrate that PA and NA do not necessarily have a completely negative relationship but are independent and can occur simultaneously (Pomerantz, Wang & Ng, 2005). Our results also support the existence of affective ambivalence (Fong, 2006)—that is, “the co-existing and intertwining positive and negative feelings toward a subject” (Pratt & Doucet, 2000; Ashforth et al., 2014; Rothman et al., 2017; Chang & Raver, 2020).

The finding that higher education teachers, who practice a highly specialized profession, had complex and varied affective experiences regarding the novel task of online teaching is also consistent with that of previous studies. In daily life, people do not reflect a single emotional state, such as simple happiness or simple sadness. Rather, they reflect mixed emotional states (Fong, 2006). For example, college students often experience strong emotional ambivalence at graduation (Larsen et al., 2004). Emotional ambivalence occurs when there is at least moderate intensity in both the PA and NA, and the PA and NA are intertwined in a similar size (Priester & Petty, 1996).

The relationship between PA, NA, and psychological adjustment of higher education teachers’ online teaching after the COVID-19 outbreak

The correlation analysis showed that PA was significantly and positively correlated with positive adjustment and significantly and negatively correlated with negative adjustment. The contrary was the case for the relationship between NA and psychological adjustment. This shows that positive affect is conducive to the psychological adjustment of online teaching work, while negative affect has the opposite on higher education teachers’ adjustment to online teaching. Like the results of this study, previous studies have also obtained that PA has a direct negative effect on maladjustment (Steptoe, Dockray & Wardle, 2009), while NA has a direct positive effect on it (Souza et al., 2008). These results validate the broaden-and-build theory. In other words, PA can broaden the immediate cognitive action resource pool, offset the influence of NA, and improve individuals’ mental resilience (Fredrickson, 2001; Kuppens, Realo & Diener, 2008).

Previous studies on changes in positive and negative affect and mental states have also indicated that PA is a major motivation for individuals to cognitions, feelings and actions that promote the broadening and building of personal and social resources (Kong & Zhao, 2013). NA is the main determinant of mental health problems, especially long-term NA. Conversely, PA reflects the pleasurable degree of individuals’ environment and corresponds to a pleasant emotional experience with direct mental health benefits (Watson & Tellegen, 1985). A previous study on the relationship between PA, NA, and mental health reported that NA leads to anxiety and pessimism (Clark & Watson, 1991). In a study with middle-aged and young people, Chang & Sanna (2007) found a significant positive correlation between PA and mental health status, while an opposite correlation with NA. Another previous study reported that life satisfaction is closely related to the experience of more positive emotions and less negative emotions (Kuppens, Realo & Diener, 2008) to adapt to changes in the environment better.

The influence of higher education teachers’ affective experience types on psychological adjustment after the COVID-19 outbreak

Meanwhile, it should be noted here that the present study is a detailed case study that adopted a person-centered perspective. The research results show that there are significant differences in the psychological adjustment statuses of higher education teachers with different affective experience types in online teaching. Higher education teachers with positive type experiences are the best at adapting to online teaching. Thus, research hypothesis H2 has been confirmed. Moreover, the analysis of variance showed some interesting results; the teachers with ambivalent type experiences had a significantly worse psychological adjustment score immediately after the COVID-19 outbreak than the teachers with common and positive type experiences. Further, the complex emotions experienced by teachers have unique and different effects on their psychological adjustment to online teaching.

As mentioned earlier, affective ambivalence refers to a mixed emotional experience in which an individual simultaneously feels positive and negative emotions. Previous studies have found that affective ambivalence has a complex impact on individuals; specifically, it has a negative impact on individuals’ interpersonal relationships in an organization (Chang & Raver, 2020). This is consistent with the results of this study. Further, there are also studies that have found that a simple emotion alone does not increase the explanatory percentage of individuals’ risky behavior. In fact, the affective ambivalence between two opposite emotions (e.g., a mixture of joy and sadness) is what causes the explanatory percentage to increase (Caballero et al., 2007). Moreover, a study of soldiers found a significant positive correlation between affective ambivalence and stress in a given situation (Jerg-Bretzke et al., 2013). Affective ambivalence is also positively correlated with guilt, loneliness, and alienation (Bruno, Lutwak & Agin, 2009).

Existing research does not adequately explain the mechanisms of how affective ambivalence influences individuals’ psychological states; it is not clear whether affective ambivalence has positive or negative effects on individuals’ development. In addition, Stratton (2015) proposed an affective events theory-based conceptual model of emotional ambivalence that describes the determinants of emotional ambivalence and their effects. This model depicts the process that individuals undergo after experiencing emotional ambivalence: paralysis, reappraisal, and finally affective-driven behavior(s). However, this conceptual model does not explicitly explain the relationship between emotional ambivalence and individuals’ behavioral patterns. Future research should therefore further explore this relationship.

When higher education teachers in the present study had to shift to the novel task and environment of online teaching immediately after the COVID-19 outbreak, many of them might have experienced NA related to the stress of the challenging situation, resulting in their negative psychological adjustment. Meanwhile, many others might have experienced the PA of relaxation brought about by the reduced supervision from their institution following the transition to online teaching. From the affect-as-information perspective, the good information from PA leads to the perception of a satisfying position, resulting in relaxed behaviors (George & Zhou, 2007). In such a scenario, teachers may tend to rely on common procedures and previous knowledge structures to cope with the work and thus not take online teaching seriously. However, when they realize that online teaching is in fact demanding and cannot be dealt with easily, they may feel nervous and anxious, which may ultimately lead to negative adjustment. Nevertheless, the relevant relationships and processes cannot yet be confidently explained. Future studies on affective ambivalence must address these issues.

Limitations and suggestions for future research

This study has some limitations. First, we based our analysis on higher education teachers’ self-evaluation, which may have been affected by social desirability. Future studies should be conducted based on more diverse data sources, including data obtained from holistic assessment. Second, we selected individual-level variables, without considering possible conflicts of interest between employees and their organization. Further investigations are needed to ascertain whether beneficial classes at the individual level are also valuable to an organization. Third, the study participants were teachers working in higher education institutions in a broad region. This may have been an advantage. However, as teachers at different levels (e.g., primary and secondary school teachers) may have a different latent profile structure, the present results may not apply to teachers at other levels or even to other professions. Fourth, although the sample size of this study is relatively large, it is a cross-sectional study after all, and the psychological adjustment characteristics of higher education teachers’ online teaching are difficult to reflect fully in an assessment. Therefore, future research can analyze the psychological adaptation characteristics and categorical characteristics of higher education teachers by adopting the longitudinal research latent variable growth model through tracking and evaluation sampling.

In addition, developmental contextualism is a perspective that emphasizes the dynamic interaction between individuals and the multiple contexts of their lives. It also incorporates the time dimension to form the circular effects model of individual development (Lerner, 2006; Sorell, SoRelle-Miner & Pausé, 2007). In other words, the relationship between PA and NA experiences and teachers’ psychological adjustment to online teaching may also have circular effects. This should be determined in future studies. Moreover, individuals’ emotions are in a state of fluctuation over time (Judge & Ilies, 2004). Dynamic changes in emotions are more representative of employees’ affective states in the actual work environment. Therefore, future research should also investigate the effects of dynamic emotions on individuals’ work quality. Finally, few studies have examined the actual interaction mechanisms between affective ambivalence and mental states. Further extensive studies are needed to understand affective ambivalence and mental states better, particularly during critical and extraordinary periods such as a pandemic.

Conclusion

This study is a case study under the complex conditions of a pandemic. From a new person-centered perspective, this study demonstrated higher education teachers’ intricate affective experiences related to online teaching during the period immediately after the COVID-19 outbreak. A novel approach was used to verify the existence of affective ambivalence. (1) Based on their affective experiences during online teaching immediately after the COVID-19 outbreak, higher education teachers were divided into three latent classes: the common, ambivalent, and positive types. (2) Among them, the positive type accounted for the largest proportion (44.85%) and the online teaching adjustment of teachers of this type was the best. Next, the common type accounted for 31.15% and their online teaching adjustment was in a normal state. An interesting finding was that while the ambivalent type accounted for the smallest proportion (23.93%), these teachers showed the worst psychological adjustment. (3) Based on relevant approaches, theories, and empirical studies on emotions, the type of affective experience in the early stage of COVID-19 has a significant impact on higher education teachers’ psychological adjustment to online teaching. In particular, affective ambivalence has a significant negative impact on the online teaching adjustment of higher education teachers. The present study extends relevant approaches, theories, and findings of empirical studies on emotions, organizational behavior, and teachers’ mental health, providing useful insights in the process.

Supplemental Information

Supplemental Information 1 All the scores and profile groupings of all subjects on the variables

Click here for additional data file.

Supplemental Information 2 The Psychological Adjustment to Online Teaching Scale (Chinese

Click here for additional data file.

Supplemental Information 3 The Psychological Adjustment to Online Teaching Scale

Click here for additional data file.

We would like to thank all the teachers who participated in this study, as well as the universities that assisted us in data collection.

Additional Information and Declarations

Competing Interests

Author Contributions

Human Ethics

Data Availability

The authors declare there are no competing interests.

Weixing Zou conceived and designed the experiments, performed the experiments, analyzed the data, prepared figures and/or tables, authored or reviewed drafts of the paper, and approved the final draft.

Xiangmei Ding and Hongli Wang conceived and designed the experiments, performed the experiments, authored or reviewed drafts of the paper, and approved the final draft.

Lingping Xie performed the experiments, analyzed the data, prepared figures and/or tables, authored or reviewed drafts of the paper, and approved the final draft.

The following information was supplied relating to ethical approvals (i.e., approving body and any reference numbers):

Ethical approval for the study was obtained from the Research Ethics Committee of the School of Education Science, Xingyi Normal University for Nationalities (No. ESXYNUN20200315).

The following information was supplied regarding data availability:

The raw measurements are available in the Supplementary File.

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
