# Peer review of "Relationship between higher education teachers’ affect and their psychological adjustment to online teaching during the COVID-19 pandemic: an application of latent profile analysis"

_PeerJ, doi:10.7717/peerj.12432_

## Round 0.1 · original submission · Major Revisions

Your manuscript was considered novel, interesting and valuable by the reviewers. However, they raised some issues that need to be addressed. Some of their concerns had to do with your methods, such as a providing a better explanation of your sampling method, explaining whether the PANAS scale was validated in the Chinese population, providing justification of the inclusion/exclusion criteria for your study population, and a more thorough description of your methodology. Other issues the reviewers raised had to do with explaining why you selected the population of University professors as well as providing more structure to the manuscript-particularly the abstract-and including a limitations section in your discussion.

Please, submit a detailed rebuttal which shows where and how you have taken all comments and suggestions into consideration. If you do not agree with some of the reviewers’ comments or suggestions, please explain why. Your rebuttal will be critical in making a final decision on your manuscript. Please, note also that your revised version may enter a new round of review by the same or by different reviewers. Therefore, I cannot guarantee that your manuscript will eventually be accepted.

·

Basic reporting

Lack of structure and clarity. Can be improved in the writing.

Experimental design

Good methodology.
Can be improved in procedure, can be improved in description of instruments

Validity of the findings

The discussion is not rich in content because it lacks structure and clarity to the reader.
The article is novel and of interest.

Additional comments

The manuscript is of interest and I consider it necessary for the authors to improve the manuscript in aspects such as the following:
-It is recommended to clarify and structure the abstract to provide clearer study conclusions.
- It is considered necessary to provide the introduction section with a greater structure and coherence. It would be appropriate to join the ideas by linking one with another.
- A further justification of the objectives of the study is needed.
-It is necessary to justify the selection of study participants, indicate reasons for inclusion, exclusion ...
- It is necessary to improve the methodology section because the procedure followed has not been clear.
- Please add sample questions from the questionnaires used.
- The discussion section add limitations.
- The discussion is not rich in content because it lacks structure and clarity to the reader.

·

Basic reporting

no comment

Experimental design

In methods, the authors, please explain more about the convenience sampling method, there remains the doubt if it is a snowball sampling or the authors chose the institutions.

The PANAS scale was validated for the Chinese population? please include the data.

Excellent statistical analysis was performed to validate the survey on psychological adjustments to online teaching.

Validity of the findings

In the discussion section, it is observed that there is little discussion about university professors. The discussion is focused on the characteristics found in the study and what could have been in any job. What makes working with university professors particular?

Additional comments

In the abstract, authors are encouraged to include the proportion of each latent class in the abstract. I think it is too ambiguous to say that one latent class is more prominent than another.

The second paragraph says, " Although China’s first large-scale online teaching program has achieved (perhaps even exceeded)," the phrase in parenthesis is not known if it is the authors' opinion or Dauguang's reference.

The introduction is engaging and complete about the issues of latent profile analysis and how affective characteristics can influence subjects. However, there remains an essential gap about university teachers, their characteristics, why to study them; university teachers were (before COVID-19) a job with risks in their health...and many more questions.

In the discussion section, it is observed that there is little discussion about university professors. The discussion is focused on the characteristics found in the study and what could have been in any job. What makes working with university professors particular?

The title of table 2 is very general. It is suggested to change the title to one related to the analysis shown.
In Table 4 and the statistical analysis section, the post-hoc test used is not indicated.

In Figure 1, I understand that these are averages. If so, the authors should include the standard deviations.

Authors are suggested to standardize the term "teacher." According to the study, a correct term should be "professor."

---

## Round 0.2 · accepted · Accept

Thank you for thoroughly addressing all the reviewers' comments, which resulted in a significant improvement of your manuscript.

·

Basic reporting

The article has improved with the revisions made

Experimental design

The article has improved with the revisions made

Validity of the findings

The article has improved with the revisions made

Additional comments

The article has improved with the revisions made